# Breaking of Sitting Time Prevents Lower Leg Swelling—Comparison among Sit, Stand and Intermittent (Sit-to-Stand Transitions) Conditions

**DOI:** 10.3390/biology11060899

**Published:** 2022-06-10

**Authors:** Rúben Francisco, Catarina L. Nunes, João Breda, Filipe Jesus, Henry Lukaski, Luís B. Sardinha, Analiza M. Silva

**Affiliations:** 1Exercise and Health Laboratory, CIPER, Faculdade Motricidade Humana, Universidade Lisboa, Estrada da Costa, 1499-002 Cruz-Quebrada, Portugal; catarinanunes@fmh.ulisboa.pt (C.L.N.); joaopmbreda@gmail.com (J.B.); filipejesus@fmh.ulisboa.pt (F.J.); lsardinha@fmh.ulisboa.pt (L.B.S.); analiza@fmh.ulisboa.pt (A.M.S.); 2Department of Kinesiology and Public Health Education, Hyslop Sports Center, University of North Dakota, Grand Forks, ND 58202, USA; henry.lukaski@und.edu

**Keywords:** bioelectrical impedance, sedentary behavior, edema, sitting, standing

## Abstract

**Simple Summary:**

It is known that staying static in the same position for a long and uninterrupted period is associated with lower limb edema. The literature shows that interruption of prolonged sitting time has a positive impact in several health-related risk factors. To determine the effects of adding sit-to-stand transitions on lower leg swelling using localized bioelectrical impedance raw parameters, we compared 3 randomized situations: (1) uninterrupted motionless standing; (2) uninterrupted motionless sitting; (3) sit-to-stand transitions (1 min sitting followed by 1 min standing). Our study demonstrates the high potential of changing from sit to stand position during short periods in preventing leg swelling.

**Abstract:**

Background: Sitting or standing during prolonged periods is related to leg swelling. It is unknown if interrupting sedentary behavior can attenuate lower leg swelling. We aimed to examine if adding sit-to-stand transitions prevents lower leg swelling as compared with uninterrupted motionless standing and sitting, using localized bioelectrical impedance raw parameters. Methods: Twenty adults participated in this crossover randomized controlled trial and acted out three conditions: (1) uninterrupted, motionless standing; (2) uninterrupted motionless sitting; (3) sit-to-stand transitions (1 min sitting followed by 1 min standing). Localized resistance (R), reactance (Xc), impedance (Z) and phase angle (PhA) were assessed at baseline, at 10 min and at 20 min for each condition. Results: For sitting and standing conditions, R and Xc values decreased after 10 and 20 min. Uninterrupted sitting resulted in the highest decrease in R (ΔSit − ΔStand = −9.5 Ω (4.0), *p* = 0.019; ΔSit − ΔInt = −11.6 Ω (4.0), *p* = 0.005). For standardized R (R/knee height), sitting was the condition with a greater decrease (ΔSit − ΔStand = −30.5 Ω/m (13.4), *p* = 0.025; ΔSit − ΔInt = −35.0 Ω/m (13.5), *p* = 0.011). Conclusions: Interrupting sedentary behavior by changing from sit to stand position during short periods may be effective at preventing leg swelling.

## 1. Introduction

An increased sedentary behavior has deleterious effects on cognitive function, mental health, physical function, disability and quality-of-life [1]. In current society, prolonged sitting time has been introduced into our days in several situations, such as transportation and in the workplace [2,3]. Additionally, some careers require employees to work in a standing position for an entire work shift without being able to walk or sit [4], such as supermarket workers [5], healthcare professionals [6] and factory/service workers [7]. Staying in the same position (i.e., sitting/standing) for a long and uninterrupted period is associated with poorer health outcomes [8,9], increasing the risk for premature mortality. More specifically, sitting for a long period of time has been related to lower limb edema [10,11]. This condition causes discomfort and pain, and is associated with a lower quality of life [12].

The literature shows that interruption of prolonged sitting time has a positive impact on several health-related risk factors, such as decreases in waist circumference, triglycerides and 2 h plasma glucose [13,14]. Moreover, a recent systematic review with meta-analysis [15] concluded that interrupting sitting time with small periods of physical activity attenuated post-prandial glucose and insulin concentrations, particularly for individuals with higher body mass index (BMI) values. Although it is known that body fluid distribution is dependent on postural changes [16,17], studies examining the effects of breaking up sitting time on the prevention of lower leg edema are scarce. Dogra and colleagues [18], using a lower leg cuff-style strain gauge, verified that hourly interruptions of sitting time by performing short bouts of 3 min of exercise on a cycle ergometer during a 4 h sitting period attenuated lower leg edema. Using circumferences to assess lower leg edema, Lin and colleagues [19] also concluded that making various leg movements caused less discomfort and swelling in participants’ lower extremities during 4 h of standing. The circumference can be used to assess the lower leg, since hydrostatic pressure occurs as a person changes the posture from standing to supine position [20].

Bioelectrical impedance analysis (BIA) is a fast, safe and non-invasive method that can also be used to determine fluid balance using whole-body measurements [21]. Raw BIA parameters such as resistance (R), reactance (Xc) and phase angle (PhA) have been associated with total body water (TBW) and its compartments (i.e., intracellular water (ICW) and extracellular water (ECW)) [22,23,24]. The R is inversely proportional to fluid volume i.e., extra- and intracellular fluids that behave as resistive components [25]. The Xc arises from cell membranes, and it describes the capacitive impedance of cell membranes [25]. Finally, PhA has been studied as an indicator of nutritional status, disease prognosis, mortality and cellular vitality [26,27], and it has been negatively correlated with fluid retention in the clinical field [28]. In terms of variations throughout the time, PhA variations were also positively associated with TBW and ICW and negatively associated with ECW:ICW ratio [29].

Usually, BIA involves a whole-body assessment, involving the use of four electrodes located on the right side (two electrodes located on the right hand and two electrodes located on the right foot) [25]. Segmental BIA has been also used over the last few years [30], as it includes assessments across body segments, i.e., the limbs and trunk [30]. Measurements may also be performed across localized regions of the body [31]. Localized BIA allows us to study changes in soft tissue hydration and cell integrity in a specific area of the body, and its use is growing [23,32]. For example, Nescolarde and colleagues [32] pointed out marked localized reductions in R, Xc and PhA caused by muscle injuries that may indicate cell membrane disruption and alteration of fluid compartments.

In the past, Codognotto and colleagues [33] provided evidence that segmental BI is a reliable indicator of alteration in the fluid status of body segments. The authors showed that the accumulation of fluid in one leg from the ankle to the knee induced a decrease in segmental impedance. However, to our knowledge only two studies verified the effects of prolonged sitting and standing time on leg swelling assessed by rate-of-change impedance using localized BIA [34,35]. Seo and colleagues [34] verified that lower leg swelling increased in ordinary chair sitting when compared with standing. In Chester’s study [35], which analyzed the effects of being seated for 90 min in an office chair vs. in a sit/stand chair, the authors indicated that sit/stand condition significantly caused the most swelling.

Additionally, no study analyzed the effects of motionless sitting with brief standing bouts on lower leg swelling. Therefore, the purpose of this study was to examine and to compare the effects of standing, sitting and sit-to-stand transitions on lower leg swelling using localized BIA measurements. We hypothesized that sit-to-stand transitions would prevent lower leg swelling when compared with uninterrupted motionless standing and sitting.

## 2. Materials and Methods

### 2.1. Study Design

Twenty healthy individuals participated in this crossover randomized experiment. All the participants performed the 3 experimental conditions with a 10 min period of laying down between them. Although 20 participants were assessed, one was not included in the analysis due to missing data. Thus, 19 participants were analyzed. The study took place at the Exercise and Health Laboratory in Faculdade Motricidade Humana, Universidade Lisboa. The study was approved by the Ethics Committee of the Faculty of Human Kinetics, University of Lisbon (Lisbon, Portugal) (CEFMH Approval Number: 7/2020) and was conducted in accordance with the declaration of Helsinki for human studies from the World Medical Association [36]. The study was registered at clinicaltrials.gov (NCT05173558).

All evaluations were performed during morning hours, starting at 7:00 am in the morning after an overnight fast.

### 2.2. Participants

The inclusion criteria were as follows: age 18–40 years old; BMI ranging between 18.5 to 29.9 kg/m^2^; not taking any medications at the time of the measurements; all women had to have a (self-reported) regular menstrual cycle (considered the non-absence of cycles throughout the year with a similar duration); and all were evaluated in the luteal phase. Participants who self-reported an inability to stand for 20 min without moving their lower bodies, with active smoking status or with presence of diabetes were excluded.

Eligible participants underwent a sequence of 3 randomly assigned experimental conditions of 20 min each. Each experimental situation was separated by 10 min rest lying in a supine position [37]: (A) 20 min of uninterrupted motionless standing, (B) 20 min of uninterrupted motionless sitting and (C) 20 min of sit-to-stand transitions (1 min sitting with 1 min standing). During the tests, participants were instructed to keep their arms extended along the trunk without movement of the lower extremities when standing, and to keep their hands on their knees without movement of the lower extremities when they were sitting. The room had an environmental temperature of 23 °C and humidity of 40–50%. The participants were asked to show up in the laboratory after overnight fasting, without consuming caffeine, to avoid performing vigorous physical activity on the previous day and not to wear metals (rings, watches, necklaces, etc.) [37].

### 2.3. Heart Rate and Blood Pressure Monitoring

The blood pressure of each participant was measured in the sitting and standing position before each experimental protocol. During the experimental condition, heart rate and blood pressure were measured every 5 min. We measured the blood pressure and heart rate as a criterion to interrupt the experiment. The experiment stopped if a 20 mm Hg or 10 mm Hg drop in systolic and/or diastolic pressure occurred [38].

### 2.4. Anthropometry

Weight and height were measured using standardized procedures [39]. The knee height (without foot height) was measured with a measuring tape, assuming the distance between the anterior surface of the thigh above the condyles of the femur and about 4 cm above the patella and the floor [40]. This measure was performed with the patient bending his/her right knee and ankle at 90° angles.

### 2.5. Dual-Energy X-ray Absorptiometry

Dual-energy X-ray absorptiometry (DXA; Hologic Explorer-W, Waltham, MA, USA) was used to determine the total fat mass (FM) and fat-free mass (FFM). A whole-body scan was performed, and the attenuation of X-rays pulsed between 70 and 140 kV synchronously with the line frequency for each pixel of the scanned image was measured. In our laboratory, in 10 healthy adults, the test–retest CV for FFM and FM were 1.1% and 1.7%, respectively.

### 2.6. Bioimpedance Analysis

A phase-sensitive bioimpedance device AKERN BIA 101/BIVA PRO was used to assess raw parameters [26]. The device measures PhA and impedance (Z) and then calculates R and Xc. The values were adjusted for height (R/H and Xc/H).

Subjects were instructed to lie in a supine position with their arms and legs abducted at 45° for 10 min [41]. After the 10 min of resting, a localized BIA measurement was taken. The subjects performed the 3 tests in random order. Between each test, a period of 10 min of rest was used, and new measurements whole body and localized BIA were taken. Additionally, at minute 10 and minute 20 of each situation, new whole body and localized measurements were taken. Low-impedance electrodes (Biatrodes, Akern Srl, Florence, Italy) were used for measuring raw parameters [42]. An alternating current was introduced into the distal electrode of each pair (source electrode), and the voltage drop across the muscle was measured using the proximal electrode (detector electrode). A whole-body test was performed by placing eight electrodes on the dorsal surfaces of both feet and ankles and at both wrists and hands (Figure 1A). For localized bioimpedance, the proximal electrode previously placed at the right wrist was removed and replaced below the inferior pole of the patella (Figure 1B) [43,44]. Participants were instructed to follow all of the bioimpedance pre-test requirements [45].

The coefficients of variation in our laboratory for repeated within-day R and Xc measures were, respectively; 1.6% and 1.9% for the right arm; 1.9% and 1.7% for the left arm; 1.9% and 1.9% for the right leg; 1.1% and 2.0% for the left leg; 1.3% and 0.1% for the right trunk; and 3.4% and 0.7% for the left trunk.

### 2.7. Statistics

Statistical analysis was performed using IBM SPSS statistics version 27.0 (IBM, Chicago, IL, USA). To test the normality of the variables, the Kolmogorov–Smirnov test was performed.

A linear mixed model was used to compare differences among the 3 conditions throughout time, adjusted for the condition order and sex as covariates and for time and condition as fixed effects. When significant findings were observed, Bonferroni post hoc testing was performed to further elucidate the differences among the 3 conditions. Contrasts were performed to calculate difference-in-differences among the 3 conditions throughout time (ex: (ΔSitbaseline − ΔSit10 min) − (ΔStandbaseline − ΔStand10 min)). For sample size calculation, this study was powered based on changes in Z. Considering a type I error of 5% and a power of 80% (using the software GPower version 3.1.9.2 (G*Power, Stuttgart, Germany)) to detect an effect size of 0.6 for statistically significant differences in impedance, as reported elsewhere [34], a total of 19 participants were required. The significance level was set at *p* < 0.05 for all tests described.

## 3. Results

Characteristics of participants are presented in Table 1.

Estimated means for localized BIA analysis, namely, R, (raw and standardized for height) Xc (raw and standardized for height), Z and PhA for sit, stand and intermittent conditions and their estimated changes are displayed in Table 2.

For sitting and standing conditions, significant decreases in R were found from baseline to 10 min and to 20 min (*p* < 0.05). The sitting condition resulted in the larger decreases in R values (ΔSit − ΔStand = −7.2 Ω (4.0), *p* = 0.047; ΔSit − ΔInt = −12.7 Ω (4.0), *p* = 0.002) when compared with standing and intermittent. No differences were found for Xc and PhA when changes were compared among conditions.

For standardized parameters, sitting was also the condition with higher decreases throughout time for R (ΔSit − ΔInt = −35.0 Ω/m (13.5), *p* = 0.011). No differences were found between standing and sit-to-stand transitions.

No differences were found among conditions for PAS or PAD throughout time (difference-in-differences, *p* > 0.05).

Mean values for R standardized for height throughout time for the three conditions are displayed at Figure 2.

## 4. Discussion

The main finding of this study was that interrupting the prolonged sitting time by transitioning from sit to stand position during short periods of time attenuated the decrease in electrical R, demonstrating a positive impact on leg swelling.

Although PhA and Xc have also been associated with TBW, ICW and ECW [22,29], R is the main raw BIA parameter associated with body fluids [25]. It is known that R is inversely proportional to fluid content [25]. Additionally, greater decreases in R after 20 min of uninterrupted sitting indicate more extracellular expansion in the lower legs, and expectedly, lower leg swelling. Therefore, it seems that interrupting this motionless state by adding short periods of sit-to-stand transitions attenuates the R decrease and consequently prevents lower leg swelling.

These results may be explained by the lower muscular activity, as a sitting posture is easy to maintain without significant muscle work. It is known that human upright positions depend upon continuous muscle activity, particularly in the legs’ muscles [46]. When comparing standing and intermittent conditions, no differences were found regarding R values, speculatively due to similar muscle activity between keeping an erect position and interrupting this motionless state by adding short periods of transition.

Codognotto and colleagues [33] demonstrated evidence that segmental BI could be a reliable indicator of alterations in fluid content from the ankle to the knee after a surgical procedure. However, the authors did not consider any movement. In another study, Seo and colleagues [34] compared 60 min of three different conditions: straight standing (where participants were allowed to walk in a small area), sitting on a buttock chair and sitting on an ordinary chair (in both conditions, subjects were permitted to do postural alterations). The authors found that sitting was the condition with the most lower leg swelling, and their findings were also likely explained by the differences in muscle activity in sitting and standing. Additionally, Chester and colleagues [35] compared 90 min of sitting, standing and sit/stand chair conditions, concluding that the use of a sit/stand chair led to a higher magnitude of lower leg swelling than the other three conditions. It should be noted that the condition sit/stand chair included the use of a chair adjusted for each participant’s height, and the seat tilted up to 12° such that the participants were semi-standing with the support of the chair. Therefore, the sit-stand chair could produce higher hydrostatic pressure, leading to higher veinous compression in the lower limbs, reducing the blood supply to the muscles and leading to blood accumulation. Nevertheless, none of these studies used an intermittent condition, where participants would be asked to realize 1 min transitions between sitting and standing position.

It was previously shown that interruptions of sitting time by performing short bouts of 3 min of exercise attenuated lower leg edema [18], and suggested that making various leg movements causes less swelling in participants’ lower extremities [19]. Congruently, our condition where participants interrupted the prolonged sitting time by transitioning from sit to stand position during short periods of time attenuated lower leg edema.

When comparing our results with previous studies, some methodological differences should be considered and lead to appropriate caution. Firstly, the duration of each condition in our study was shorter when compared to other studies [34,35]. We wanted to demonstrate that even for shorter periods, interrupting static positions may be important. Additionally, the amount of movement allowed throughout the experiment is also an important factor that could influence the results. In the aforementioned studies [34,35], minor lower legs movements and postural changes were allowed. On the other hand, in our study, participants were instructed not to move their lower extremities and to keep their arms extended along the trunk when standing, and to keep their hands on their knees without movement of the lower extremities when they were sitting.

Lastly, to our knowledge, only two studies examined the dynamics of lower leg swelling in sitting and standing conditions using localized BIA [34,35]. Nevertheless, neither studied the effects of interrupting a prolonged motionless condition by performing sit-to-stand transitions on lower leg swelling. Thus, our results are useful in a work-field environment, especially for those who need to work in static positions and could benefit from disrupting those positions, such as supermarket workers [5], healthcare professionals [6] and factory and service workers [7]. Although our findings are limited only to 20 min and most real situations, such as work shifts, require more time, this study highlights a very important message: that even for shorter periods, small movements may be important for reducing leg swelling.

The limitations of the study need to be addressed. First, as our population comprised 18–40 year-old adults, our findings cannot be extrapolated to other populations, such as a younger/older sample or populations with specific diseases/conditions. Additionally, although we instructed the participants to avoid any lower leg movement, and two researchers closely monitored each condition, we were not able to assure that no movement occurred. Finally, this study presented acute effects, but chronic effects still require additional research. Therefore, it remains to be determined whether interrupting the prolonged sitting time will prevent lower body edema in the long term.

## 5. Conclusions

In conclusion, interrupting the prolonged sitting time by transitioning from sitting to standing position during short periods of time attenuates the decrease in electrical R, likely avoiding extracellular water expansion, ultimately preventing lower leg swelling. Our data may be helpful for those who need to stay in static positions for long periods of time. Although our findings are limited only to the time of the studied conditions, this study brings a very important message that even for shorter periods, small movements may be important for reducing leg swelling. Greater lower leg edema is likely when participants remain seated.

## Figures and Tables

**Figure 1 biology-11-00899-f001:**
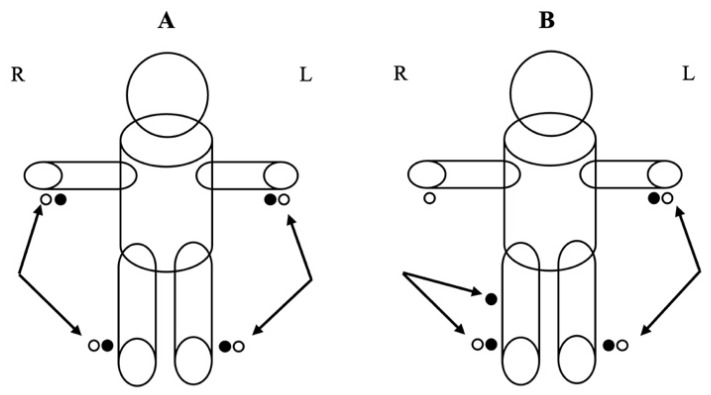
Depicture of whole-body analysis (**A**) and localized analysis (**B**). (R—right side, L—left side). The filled balls represent the voltage electrodes, and the blank balls represent the current electrodes.

**Figure 2 biology-11-00899-f002:**
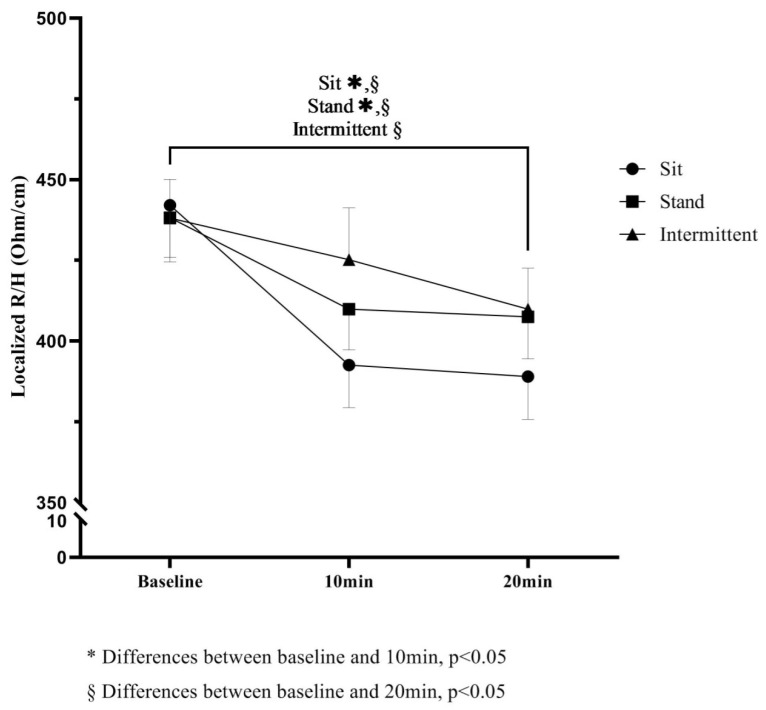
Localized R values standardized for height throughout time.

**Table 1 biology-11-00899-t001:** Characteristics of participants.

	Total Sample (*n* = 19)	Females (*n* = 9)	Males (*n* = 10)
Demographic			
Age	27.5 ± 5.9	29.3 ± 7.4	25.8 ± 3.9
Body Composition			
Weight (kg)	70.6 ± 12.0	66.2 ± 13.3	75.6 ± 10.0
BMI (kg/m^2^)	24.3 ± 3.6	24.1 ± 4.8	24.4 ± 2.3
Fat mass (kg)	17.4 ± 8.9	21.5 ± 11.1	13.7 ± 4.1
Fat mass (%)	24.5 ± 10.1	31.2 ± 10.6	18.6 ± 4.1
Fat-free mass (kg)	52.3 ± 10.2	43.8 ± 3.2	60.0 ± 7.8
Blood Pressure			
SBP (mmHg)	118.8 ± 9.9	113.4 ± 10.0	124.1 ± 6.7
DBP (mmHg)	70.4 ± 10.0	68.3 ± 6.0	72.6 ± 13.0
Whole-Body Bioimpedance			
R (Ω)	532.9 ± 116.7	614.8 ± 117.9	459.3 ± 46.6
Xc (Ω)	65.3 ± 14.9	67.5 ± 19.7	63.3 ± 9.4
PhA (°)	7.1 ± 1.3	6.3 ± 1.1	7.9 ± 1.0
TBW (L)	35.3 ± 7.6	28.8 ± 4.0	41.2 ± 4.5
ECW (L)	15.4 ± 3.0	13.0 ± 1.8	17.5 ± 2.2
ICW (L)	20.0 ± 4.6	15.8 ± 2.2	23.7 ± 2.3

Data are presented as mean ± SD. BMI, body mass index. SBP, systolic blood pressure. DBP, diastolic blood pressure. R, resistance. Xc, reactance. PhA, phase angle. TBW, total body water. ECW, extracellular water. ICW, intracellular water.

**Table 2 biology-11-00899-t002:** Values for localized BIA parameters for sit, stand and intermittent conditions and difference-in-differences among conditions (linear mixed models, adjusted for the condition’s order and sex).

		Sit	Stand	Intermittent	Difference-in-Differences		
R_localized_	Baseline	158.1 (4.0)	157.4 (4.0)	156.5 (4.0)	ΔSit − ΔStand	ΔSit − ΔInt	ΔStand − ΔInt
	10 min	141.1 (4.0) ^†^	147.6 (4.0) ^†^	152.1 (4.0)	−7.2 (4.0) *	−12.7 (4.0) *	−5.5 (4.0)
	20 min	137.6 (4.0) ^‡^	146.4 (4.0) ^‡^	147.5 (4.0)	−9.5 (4.0) *	−11.6 (4.0) *	−2.0 (4.1)
Xc_localized_	Baseline	16.8 (1.0)	12.7 (1.0)	13.2 (1.0)	ΔSit − ΔStand	ΔSit − ΔInt	ΔStand − ΔInt
	10 min	15.5 (1.0) ^†^	12.8 (1.0)	12.5 (1.0)	−1.3 (1.8)	−1.0 (1.7)	0.4 (1.8)
	20 min	17.2 (1.0)	14.1 (1.0)	12.2 (1.0) ^‡^	−0.6 (1.8)	1.4 (1.8)	2.0 (1.8)
Z_localized_	Baseline	159.0 (4.6)	141.6 (4.6)	131.7 (4.6)	ΔSit − ΔStand	ΔSit − ΔInt	ΔStand − ΔInt
	10 min	158.1 (4.7) ^†^	148.1 (4.7)	146.9 (4.7)	−7.3 (6.0)	−12.8 (6.0) *	−5.5 (6.1)
	20 min	157.4 (4.7) ^‡^	152.8 (4.6)	148.1 (4.7)	−16.0 (6.0) *	−18.0 (6.1) *	−1.9 (6.2)
PhA_localized_	Baseline	6.1 (0.4)	5.7 (0.4)	6.0 (0.4)	ΔSit − ΔStand	ΔSit − ΔInt	ΔStand − ΔInt
	10 min	5.2 (0.4)	5.0 (0.4)	5.2 (0.4)	−0.2 (0.5)	−0.2 (0.5)	<−0.1 (0.5)
	20 min	5.5 (0.4)	4.9 (0.3)	4.8 (0.4)	0.3 (0.5)	0.6 (0.5)	0.3 (0.5)
R/H_localized_	Baseline	443.5 (13.8)	439.8 (13.9)	436.1 (14.0)	ΔSit − ΔStand	ΔSit − ΔInt	ΔStand − ΔInt
	10 min	393.1 (13.8) ^†^	411.1 (13.8)	426.1 (13.8)	−21.7 (13.3)	−40.5 (13.4) *	−18.7 (13.5)
	20 min	383.2 (13.9) ^‡^	410.1 (13.8)	410.8 (13.8)	−30.5 (13.4) *	−35.0 (13.5) *	−4.5 (13.5)
Xc/H_localized_	Baseline	47.8 (3.4)	42.7 (3.4)	48.9 (3.4)	ΔSit − ΔStand	ΔSit − ΔInt	ΔStand − ΔInt
	10 min	35.4 (3.3)	34.9 (3.3)	38.9 (3.3)	−4.6 (5.7)	−2.4 (5.7)	−2.2 (5.8)
	20 min	35.8 (3.4)	35.8 (3.4)	33.1 (3.3) ^‡^	−5.1 (5.8)	3.7 (5.7)	8.9 (5.8)

Data are estimated means and standard error (SE). * Difference-in-differences among conditions, *p* < 0.05. ^†^ Differences between baseline and 10 min, *p* < 0.05. ^‡^ Differences between baseline and 20 min, *p* < 0.05. R/H, resistance standardized for height; Xc/H, reactance standardized for height; H, height.

## Data Availability

The data that support the findings of this study are available from the corresponding author upon reasonable request.

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
