# Peer review of "Breaking of Sitting Time Prevents Lower Leg Swelling—Comparison among Sit, Stand and Intermittent (Sit-to-Stand Transitions) Conditions"

_biology, 2022, doi:10.3390/biology11060899_

Round 1

Reviewer 1 Report

 Breaking of sitting time prevents lower leg swelling - comparison among sit, stand and intermittent (sit-to-stand transitions) conditions.

Thank you for giving me the opportunity to review this manuscript. The main question here is whether a sit-to-stand transition may cause a lower leg swelling compared to standing and sitting alone. Sit-to-stand seems to favours lower leg swelling and this finding might have many applications to the real world. Authors have done a great work here and I would like to comment their effort. I have only some minor comments, and then I believe that this study would be ready for publication.

Overall the manuscript is well written and provides a good and analytical presentation of the scientific literature. Authors present the advantages of BIA using it as a diagnose system with many variables. I totally agree with authors that sitting or standing for a long period of time should be interrupted with movement. The study is well organized, however, methods and results needs revisions.

Line 103: “A crossover randomized experiment was performed among 20 healthy individuals”. Is that means that all 20 participants (19 participants at the end) followed all 3 conditions? Consequently, the statistical analysis included N=19 participants for each experimental condition?

Lines 120-124: Why 20 minutes were chosen here?

Line 121: What was the actual position of the 10 minutes rest? I made the connection only after reaching the lines 165-166.  I suggest providing a small description here for readers.

Lines 150-156: Why dual-energy x-ray absorptiometry was performed? Is there a correlation between fat mass and leg swelling? Is there a connection between fat free mass and leg swelling?

Authors provide CV% for DEXA measurement but failed to provide CV% for all other measurements. Please, add CV% for all measurements especially for BIA.

Lines 175-178: Although I agree with Authors for this technique, is there a reference here to support the movement of the electrodes?

In the results section delete the points referred to in the placement of the tables and figures.

Table 1 is good. Please, add more details into the legend of Table 1.

Table 2 needs a legend. Please, add a specific description above.

Figure 1 is very informative, well done here.

Figure 2 is also very good. I would recommend enlarge it a little.

Minor comments:

Line 75: Please describe ICW before using the abbreviation.

Lines 86 and 255: Is this segment BIA?

Line 111: Please change “in one moment” to “during morning hours”.

Line 152: Fat free mass or lean mass? What was the actual result from DEXA?

Line 267: “12o” is not clear. Please provide more details.

Author Response

Thank you for giving me the opportunity to review this manuscript. The main question here is whether a sit-to-stand transition may cause a lower leg swelling compared to standing and sitting alone. Sit-to-stand seems to favours lower leg swelling and this finding might have many applications to the real world. Authors have done a great work here and I would like to comment their effort. I have only some minor comments, and then I believe that this study would be ready for publication.

Overall the manuscript is well written and provides a good and analytical presentation of the scientific literature. Authors present the advantages of BIA using it as a diagnose system with many variables. I totally agree with authors that sitting or standing for a long period of time should be interrupted with movement. The study is well organized, however, methods and results needs revisions.

R: Dear reviewer,

The authors of this manuscript appreciate your review and comments. All the comments were considered for improving our work. Thank you. All the changes are marked as “track mode”.

Line 103: “A crossover randomized experiment was performed among 20 healthy individuals”. Is that means that all 20 participants (19 participants at the end) followed all 3 conditions? Consequently, the statistical analysis included N=19 participants for each experimental condition?

R: Thank you for the question. Yes, 20 participants were initially included, however, only 19 were analyzed due to missing data. All the participants performed the 3 experimental conditions with a 10 min period of laying down between them. We added information in the regard in the study design section.

Lines 120-124: Why 20 minutes were chosen here?

R: Previously, Seo and colleagues (Seo et al., 1996) performed a similar experiment using a 60min protocol. Also, Chester and colleagues (Chester et al., 2002) chose a 90min protocol. First, our purpose was to demonstrate that even for smaller periods of time, interrupting the sedentary time could be essential to observe differences in leg swelling. Second, in Seo’s study, the authors indicated differences from 30 min on but a slight tendency starting at about minute 20. The authors analyzed 12 participants (eight males and four females); thus the reduced number of participants may have limited the power to detect differences within and between situations (sit, stand and sit/stand). Thus, we knew that even smaller periods of time may be interesting to analyze. 

We think this study brings an important message that even in short periods of time, the interruption of sedentary time may be significant.

We added some information about the protocol choice in the discussion section. Also, we did some changes to the practical message of this study. In our opinion, the fact of the protocol is 20min brings an important message. Interrupting sedentary time may be important for those who spend long periods in static positions but even for those who are exposed to short bouts of static position, interrupting the static positions may be important. We discuss this point in the discussion section and also in the conclusion section.

Seo et al. 1996 DOI https://doi.org/10.1539/joh.38.186

Chester et al. DOI https://doi.org/10.1016/S0169-8141(01)00069-5

Line 121: What was the actual position of the 10 minutes rest? I made the connection only after reaching the lines 165-166.  I suggest providing a small description here for readers.

R: Thank you. The information “Each experimental situation was separated by 10min rest lying in a supine position” was added and a citation of Kyle et al., 2004 as well.

Kyle et al., 2004 DOI: 10.1016/j.clnu.2004.09.012

Lines 150-156: Why dual-energy x-ray absorptiometry was performed? Is there a correlation between fat mass and leg swelling? Is there a connection between fat free mass and leg swelling?

Authors provide CV% for DEXA measurement but failed to provide CV% for all other measurements. Please, add CV% for all measurements especially for BIA.

R: The information about dual-energy x-ray was used to characterize the participants. We did not study any relation between body composition and leg swelling. Regarding the CV, we added information for BIA. We would kindly ask you to accept another change in the BIA section. We had to change the electrodes information because we used the electrodes “biatrodes, Akern Srl” and not the Impedimed electrodes.  We apologize for the mistake.

Lines 175-178: Although I agree with Authors for this technique, is there a reference here to support the movement of the electrodes?

R: The supportive reference for this technique was Bogónez-Franco et al. (2015). In that study, we can find a detailed description of how to use the localized and segmental BIA. Also, we added Seward B. Rutkove’s work that we consider supportive information for the technique.

The papers can be seen using the following DOI:

Rutkove:  https://doi.org/10.1002/mus.21362

Bogónez-Franco et al. doi:10.1088/0967-3334/36/1/85 

We added the reference in the text. Thank you.

In the results section delete the points referred to in the placement of the tables and figures.

R: Thank you. The points were deleted.

Table 1 is good. Please, add more details into the legend of Table 1.

R: “BMI, body mass index. SBP, systolic blood pressure. DBP, diastolic blood pressure. R, resistance. Xc, reactance. PhA, phase angle. TBW, total body water. ECW, extracellular water. ICW, intracellular water.” This information was added.

Table 2 needs a legend. Please, add a specific description above.

R: We apologize for the mistake. We lost the legend while we were inserting the manuscript in the journal’s template. 

Figure 1 is very informative, well done here.

R: Thank you.

Figure 2 is also very good. I would recommend enlarge it a little.

R: Thank you. The figure was modified according to the reviewer’s comment.

Minor comments:

Line 75: Please describe ICW before using the abbreviation.

R: The information was added in introduction section. Also, the authors included the same information for ECW.

Lines 86 and 255: Is this segment BIA?

In Codognotto’s study, they report the following sentence: “Segmental Z from the lower limbs (Z-leg) was measured by leaving on each foot the same pair of electrodes as for Z-body and placing the other pair on the trochanteric region on the same side of the body. The detector electrode was placed on the greater trochanter, and the source electrode was placed 7 cm proximally on the thigh”. Thus, we wrote segmental in respect for the authors’ work. As we can see in the following paper:  DOI: 10.1088/0967-3334/36/1/85 , the electrodes placement of Codognotto’s study is performed based on segmental placement guidelines.

Line 111: Please change “in one moment” to “during morning hours”.

R: Thank you. We changed the sentence.

Line 152: Fat free mass or lean mass? What was the actual result from DEXA?

R: DXA allows both regional and total body composition estimates, characterizing fat mass (FM) and dividing fat-free mass (FFM) into two components, lean soft tissue (LST) and bone mineral content (BMC). In this current study, we mention the whole body FFM and FM. We added the CV for FFM rather than the LST CV.

Line 267: “12o” is not clear. Please provide more details

R: Thank you. We changed it from “12o” to “12o

Reviewer 2 Report

Methodology. Line 232. Title Table 2. It is advisable to revise the wording. 

Measurement protocol. It is considered that the protocol does not adjust to the reality of occupational exposure, and its relationship with possible rest periods. It would be more transferable to the workplace if rest periods were combined between longer work shifts. It is recommended that information be provided to justify the choice of protocol (times, periods, duration, etc.).

Author Response

R: Dear reviewer,

Thank you for your comments. They were useful, and we think that the overall quality was improved due to your comments, especially the second one related to the time chosen for the experiment.

Methodology. Line 232. Title Table 2. It is advisable to revise the wording. 

R: We apologize for the mistake. We lost the legend while we were inserting the manuscript in the journal’s template.  We revised the wording as recommended.

Measurement protocol. It is considered that the protocol does not adjust to the reality of occupational exposure, and its relationship with possible rest periods. It would be more transferable to the workplace if rest periods were combined between longer work shifts. It is recommended that information be provided to justify the choice of protocol (times, periods, duration, etc.).

R: Previously, Seo and colleagues (Seo et al., 1996) performed a similar experiment using a 60min protocol. Also, Chester and colleagues (Chester et al., 2002) chose a 90min protocol. First, our purpose was to demonstrate that even for smaller periods of time, interrupting the sedentary time could be essential to observe differences in leg swelling. Second, in Seo’s study, the authors indicated differences from 30 min on but a slight tendency starting at about minute 20. The authors analyzed 12 participants (eight males and four females), thus the reduced number of participants may have limited the power to detect differences within and between situations (sit, stand and sit/stand). Thus, we knew that even smaller periods of time may be interesting to analyze.

Seo et al. 1996 DOI https://doi.org/10.1539/joh.38.186

Chester et al. DOI https://doi.org/10.1016/S0169-8141(01)00069-5

We added some information about the protocol choice in the discussion section. Also, we did some changes to the practical message of this study. In our opinion, the fact of the protocol is 20min brings an important message. Interrupting sedentary time may be important for those who spend long periods in static positions but even for those who are exposed to short bouts of static position, interrupting the static positions may be important. We discuss this point in the discussion section.

Reviewer 3 Report

The authors are to be commended for their efforts in conducting the investigation of intermittent sit-to-stand transitions in effecting changes in lower leg swelling. The decision of this reviewer is based on the following evidence.

Major Concerns

1. General lack of oversight and attention. For example, three different sample sizes were reported (N = 21 in Abstract; N = 20 in Methods; N = 19 in Table 1). Can you clarify which is the correct sample size being used in your reporting of the results? Another example is your research question. You hypothesize an effect between sit and intermittent conditions but your title suggests a comparison between prolonged standing and intermittent conditions. Is there a reason for not developing a similar hypothesis for this comparison? In keeping with the hypothesis, what do you mean by muscle pump? And because of some of your findings, should it not be discussed more (in greater detail) in the Discussion?

2. Presentation of the manuscript. For example, Figure 2 is difficult to read due to the small size. Table 2 description needs to be addressed. Likewise, legend at bottom of Table 2 needs to be corrected (e.g., standard deviations vs. SE / line 233). Authors do not provide any description or detail of the findings in the Results section. Rather, they seem to allow the tables and figures to describe the findings without providing any context.

3. Again, no description of the muscle pump, although important enough to include in hypothesis.

3. Rewrite 12 degree with superscript degree symbol (line 267).

4. Change from sit to seated (Conclusions / line 311).

5. References. Please address the inconsistencies found in the formatting of all references (e.g., journal title [multiple], missing page numbers [3], missing volume [21]).  

Minor Concerns

1. Consider Keyword choices. Edema, sedentary behavior, bioimpedance seem appropriate. What about resistance and fluid distribution or total body water? Current others seem questionable.

2. Abbreviate extra- and intracellular fluids after spell out (line 70).

3. Word choice on line 94 (higher lower vs greater lower?).

4. What was significance level set at (Statistics / line 180)?

Author Response

Dear reviewer,

The authors are grateful for the detailed and in-depth review. Your comments were extremely helpful and highlighted some errors that are in our interest to correct.

Major Concerns

  1. General lack of oversight and attention. For example, three different sample sizes were reported (N = 21 in Abstract; N = 20 in Methods; N = 19 in Table 1). Can you clarify which is the correct sample size being used in your reporting of the results? Another example is your research question. You hypothesize an effect between sit and intermittent conditions but your title suggests a comparison between prolonged standing and intermittent conditions. Is there a reason for not developing a similar hypothesis for this comparison? In keeping with the hypothesis, what do you mean by muscle pump? And because of some of your findings, should it not be discussed more (in greater detail) in the Discussion?

Thank you for the comment. The authors apologize for the mistake in the sample size in the abstract. We have corrected the number in the abstract. We recruited 20 participants. Then, one of them was not included in the analysis due to missing data. It means that 20 subjects participated in our study and performed the 3 experimental situations but then we analyzed 19 subjects. To better clarify the readers of this manuscript, we also included a correction in the methods section.

The authors had some difficulty in understanding the point of view of the following topic: "You hypothesize an effect between sit and intermittent conditions, but your title suggests a comparison between prolonged standing and intermittent conditions. I there a reason for not developing a similar hypothesis for this comparison?” We consider that the title is clear. We compared 3 situations (sit, stand and intermittent). We did minor changes in the last paragraph of the introduction for better clarification of our hypothesis. However, it is in our interest to improve the manuscript according to the reviewer's opinion. So, if the reviewer considers we need to do more changes we would like to have a better clarification of the mentioned topic.

We deleted the muscle pump from our hypothesis considering that we are not able to assess the muscle pump. When we formulated the hypothesis, we thought that sit-stand transitions may be useful in preventing leg swelling due to the muscle pump (contractions). Since we are not able to assess the muscle pump, we deleted this information.

  1. Presentation of the manuscript. For example, Figure 2 is difficult to read due to the small size. Table 2 description needs to be addressed. Likewise, legend at bottom of Table 2 needs to be corrected (e.g., standard deviations vs. SE / line 233). Authors do not provide any description or detail of the findings in the Results section. Rather, they seem to allow the tables and figures to describe the findings without providing any context.

R: Thank you. The figure 2 was modified according to the reviewer’s comment. We also added a description for table 2 and corrections for the information at bottom of table 2. We apologize for the mistake. We lost the legend while we were inserting the manuscript into the journal’s template.

Regarding the results section, we have a descriptive table 1 with the participants' characteristics. The table explains itself with no need for a repetition of information. The main results are in both text and table 2. Once again, it is our interest to understand and follow the reviewers’ comments for an overall quality improvement. Thus, we would be grateful for a better clarification about this topic and what we can change to improve this section.

  1. Again, no description of the muscle pump, although important enough to include in hypothesis.

R: Based on previous comments, we deleted the term “muscle pump”. We found that it is not the most correct term.

  1. Rewrite 12 degree with superscript degree symbol (line 267).

R: Thank you. We corrected this mistake.

  1. Change from sit to seated (Conclusions / line 311).

R: Thank you. We did this change.

  1. References. Please address the inconsistencies found in the formatting of all references (e.g., journal title [multiple], missing page numbers [3], missing volume [21]).  

R: Thank you. We did the mentioned changes in the references section. We also did an update of all references using the Endnote software as recommended in the “instructions for authors” in the Biology main page.

Minor Concerns

  1. Consider Keyword choices. Edema, sedentary behavior, bioimpedance seem appropriate. What about resistance and fluid distribution or total body water? Current others seem questionable.

R: Thank you for the suggestion. We did changes to the keywords. The changes are in track mode in the manuscript. Thank you.

  1. Abbreviate extra- and intracellular fluids after spell out (line 70).

R: Thank you. We added the abbreviations in the mentioned line.

  1. Word choice on line 94 (higher lower vs greater lower?).

R: We changed the sentence using other words. Thank you for the suggestion.

  1. What was significance level set at (Statistics / line 180)?

R: We added information about the significance level in the statistics. Thank you.

Round 2

Reviewer 1 Report

no comments

Reviewer 3 Report

The authors are to be commended for addressing the issues raised by this reviewer. In doing so, the manuscript and its content have been improved.